# GENERATIVE ENSEMBLES FOR ROBUST ANOMALY DETECTION

## ABSTRACT

Deep generative models are capable of learning probability distributions over large, high-dimensional datasets such as images, video and natural language. Generative models trained on samples from $p(\mathrm{x})$ ought to assign low likelihoods to out-of-distribution (OoD) samples from $q(\mathrm{x})$, making them suitable for anomaly detection applications. We show that in practice, likelihood models are themselves susceptible to OoD errors, and even assign large likelihoods to images from other natural datasets. To mitigate these issues, we propose Generative Ensembles, a model-independent technique for OoD detection that combines density-based anomaly detection with uncertainty estimation. Our method outperforms the Out-of-DIstribution detector for Neural networks (ODIN) and Variational Information Bottleneck (VIB) baselines on image datasets, and achieves comparable performance to a classification model on the Kaggle Credit Fraud dataset.

## 1 INTRODUCTION

Knowing when a machine learning (ML) model is qualified to make predictions on an input is critical to safe deployment of ML technology in the real world. When training and test distributions differ, neural networks may provide – with high confidence – arbitrary predictions on inputs that they are unaccustomed to seeing. To mitigate these Out-of-Distribution (OoD) errors, we require methods to determine whether a given input is sampled from a different stochastic generator than the one used to train the model.

OoD detection techniques have broad applications beyond safe deployment of ML technology. As datasets for ML grow ever larger and trend towards automated data collection, we require scalable methods for identifying outliers and quantifying noise before we can attempt to train models on that data. Identifying anomalies in data is a crucial feature of many data-driven applications, such as credit fraud detection and monitoring patient data in medical settings.

Generative modeling algorithms have improved dramatically in recent years, and are capable of learning probabilistic models over large, high-dimensional datasets such as images, video, and natural language (Vaswani et al., 2017; Wang et al., 2018). A generative model $p_\theta(\mathrm{x})$, parameterized by random variable $\theta$ and trained to approximate data distribution $p(\mathrm{x})$, ought to assign low likelihoods to samples from any distribution $q(\mathrm{x})$ that differs from $p(\mathrm{x})$. Density estimation does not presuppose a specific "alternate" distribution at training time, making it an attractive alternative to classification-based anomaly detection methods.

In this work, we apply several classes of generative models to OoD detection problems and demonstrate a significant shortcoming to high-dimensional density estimation models: *the anomaly detection model itself may be mispecified*. Explicit likelihood models can, in practice, realize high likelihoods to adversarial examples, random noise, and even other natural image datasets. We also illustrate how GAN discriminators presuppose a particular OoD distribution, which makes them particularly fragile at OoD classification. We propose Generative Ensembles, which combine density estimation with uncertainty estimation to detect OoD in a robust manner. Generative Ensembles are model-independent and are trained independently of the task-specific ML model of interest. Our method outperforms task-specific OoD baselines on the majority of evaluated OoD tasks and demonstrate competitive results with discriminative classification approaches on the Kaggle Credit Fraud dataset.

## 2 GENERATIVE ENSEMBLES

We consider several classes of generative modeling techniques in our experiments. Autoregressive Models and Normalizing Flows (NF) are fully-observed likelihood models that construct a tractable log-likelihood approximation to the data-generating density $p(\mathrm{x})$ (Uria et al., 2016; Dinh et al., 2014; Rezende & Mohamed, 2015). Variational Autoencoders (VAE) are latent variable models that maximize a variational lower bound on log density (Kingma & Welling, 2013; Rezende et al., 2014). Finally, Generative Adversarial Networks (GAN) are implicit density models that minimize a divergence metric between $p(\mathrm{x})$ and generative distribution $q_\theta(\mathrm{x})$ (Goodfellow et al., 2014). We refer to a GAN's generative distribution as $q_\theta(\mathrm{x})$ (in lieu of $p_\theta(\mathrm{x})$) because from the GAN discriminator's point of view, the outputs of the generator are OoD and depend on $\theta$.

Although $\log p(\mathrm{x})$ and its lower bounds are proper scoring methods (Lakshminarayanan et al., 2017), we approximate them in practice with continuous-valued neural network function approximators $\log p_\theta(\mathrm{x})$. Neural networks have non-smooth predictive distributions, which makes them susceptible to malformed inputs that exploit idiosyncratic computation within the model (Szegedy et al., 2013).

Likelihood function approximators are no exception. When judging natural images, we assume an OoD input $x \sim q(\mathrm{x})$ should remain OoD within some $L^P$-norm, and yet a Fast Gradient Sign Method (FGSM) attack (Goodfellow et al., 2015) on the predictive distribution can realize extremely high likelihood predictions (Nguyen et al., 2015). Conversely, a FGSM attack in the reverse direction on an in-distribution sample $x \sim p(\mathrm{x})$ creates a perceptually identical input with low likelihood predictions (Kos et al., 2018). To make matters worse, we show in Figure 1 that likelihood models can be fooled by OoD samples that are not even adversarial by construction, such as SVHN test images on a likelihood model trained on CIFAR-10. Concurrent work by Nalisnick et al. (2018) also show this phenomena and present additional analyses on why generative models systematically assign higher likelihoods to SVHN.

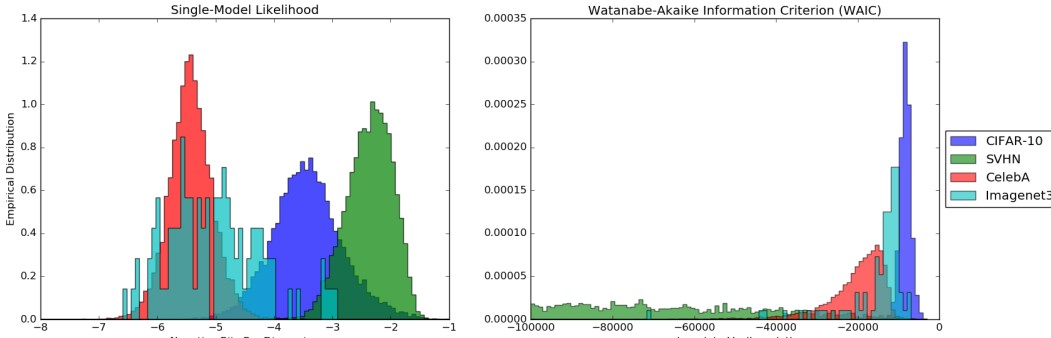

Figure 1: Left: density estimation models are not robust to OoD inputs. A GLOW model (Kingma & Dhariwal, 2018) trained on CIFAR-10 assigns much higher likelihoods to samples from SVHN than samples from CIFAR-10. Right: We use ensembles of generative models to implement the Watanabe-Akaike Information Criterion (WAIC), which combines density estimation with uncertainty estimation. Histograms correspond to predictions over test sets from each dataset.

Generative Ensembles detect OoD examples by combining a density evaluation model with predictive uncertainty estimation on the density model via ensemble variance. Following the results of Lakshminarayanan et al. (2017), we elect to use independently trained ensembles instead of a Bayesian Dropout approximation (Gal & Ghahramani, 2016). For generative models that admit exact likelihoods (or variational approximations), the ensemble can be used to implement the Watanabe-Akaike Information Criterion (WAIC), which consists of a density estimation score with a Bayesian correction term for model bias (Watanabe, 2010):

$$\mathrm{WAIC}(x) = \mathbb{E}_\theta[\log p_\theta(\mathrm{x})] - \mathrm{Var}_\theta\left[\log p_\theta(\mathrm{x})\right] \tag{1}$$

## 2.1 OoD Detection with GAN Discriminators

We describe how to construct Generative Ensembles based on implicit density models such as GANs, and highlight the importance of OoD detection approaches that do not presuppose a specific OoD distribution. A discriminative model tasked with classifying between $p(\mathrm{x})$ and $q(\mathrm{x})$ is fragile to inputs that lie in neither distribution. Figure 2b illustrates a simple 2D density modeling task where individual GAN discriminators – when trained to convergence – learn a discriminative boundary that does not adequately capture $p(\mathrm{x})$.

However, *unlike* discriminative anomaly classifiers on a static datasets, which model $p(\mathrm{x})/q(\mathrm{x})$, the likelihood ratio $p(\mathrm{x})/q_\theta(\mathrm{x})$ implicitly assumed by a GAN discriminator is uniquely randomized by GAN training dynamics on $\theta$. By training an ensemble of GANs we can estimate the posterior distribution over model decision boundaries $p(\mathrm{x})/q_\theta(\mathrm{x})$, or equivalently, the posterior distribution over alternate distributions $q_\theta(\mathrm{x})$. In other words, we can use uncertainty estimation on randomly sampled discriminators to de-correlate the OoD classification errors made by a single discriminator (Figure 2c).

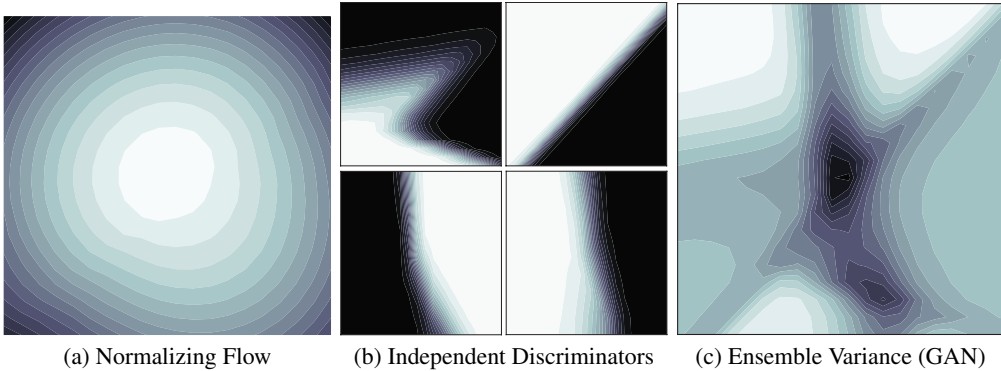

(a) Normalizing Flow      (b) Independent Discriminators      (c) Ensemble Variance (GAN)

Figure 2: In this toy example, we learn generative models for a 2D multivariate normal with identity covariance centered at (5, 5). (a) Explicit density models such as Normalizing Flows concentrate probability mass at the data distribution (b) Four independently trained GANs learn random discriminative boundaries, each corresponding to a different implied generator distribution. To ensure that the GAN discriminators form a clear discriminative boundary between $p(\mathrm{x})$ and $q_\theta(\mathrm{x})$, we train the discriminators an additional 10k steps to convergence. Each of these boundaries fails to enclose the true data distribution. (c) Predictive uncertainty over an ensemble of discriminators "fences in" the shared, low-variance region corresponding to $p(\mathrm{x})$.

## 3 Related Work

We can categorize existing OoD detection techniques in Table 1 using two criteria: (1) Does it assume a specific anomaly distribution? (2) Is the technique specific to the model, or does it only depend on the inputs to the model?

A common approach to OoD detection (a.k.a. anomaly detection) is to label a dataset of anomalous data and train a binary classifier on that label. Alternatively, a classification task model may be augmented with a "None of the above" class. The classifier then learns a decision boundary (likelihood ratio) between $p(\mathrm{x})$ and $q(\mathrm{x})$. However, the discriminative approach to anomaly detection requires the anomaly distribution to be specified at training time; this is a severe flaw when anomalous data is rare (e.g. medical seizures) or non-stationary (e.g. generated by an adversary).

### 3.1 Uncertainty Estimation

OoD detection is closely related to the problem of uncertainty estimation, whose goal is to yield calibrated confidence measures for a model's predictive distribution $p_\theta(\mathrm{y}|\mathrm{x})$. Well-calibrated uncertainty estimation integrates several forms of uncertainty into $p_\theta(\mathrm{y}|\mathrm{x})$: model misspecification un-

Table 1: Categorization of several OoD detection techniques, based on whether they depend on a specific model/task, and whether they assume a specific anomaly distribution.

|  | Model-Dependent | Model-Independent |
|---|---|---|
| OoD Dependent | Auxiliary "Other" class | Binary classification (likelihood ratio) Adversarial Training |
| OoD Independent | Hendrycks & Gimpel (2016) Gal & Ghahramani (2016) Liang et al. (2017) Lakshminarayanan et al. (2017) Alemi et al. (2018b) | Density Estimation Generative Ensembles (ours) |

certainty (OoD detection of invalid inputs), aleatoric uncertainty (irreducible input noise for valid inputs), and epistemic uncertainty (unknown model parameters for valid inputs). In this paper, we study OoD detection in isolation; instead of considering whether $p_\theta(\mathrm{y}|\mathrm{x})$ should be trusted for a given $x$, we are trying to determine whether $x$ should be fed into $p_\theta(\mathrm{y}|\mathrm{x})$ at all.

Predictive uncertainty estimation is a model-dependent OoD technique because it depends on task-specific information (such as labels and task model architecture) in order to yield an integrated estimate of uncertainty. ODIN (Liang et al., 2017), MC Dropout (Gal & Ghahramani, 2016) and DeepEnsemble (Lakshminarayanan et al., 2017) model a calibrated predictive distribution for a classification task. Variational information bottleneck (VIB) (Alemi et al., 2018b) performs divergence estimation in latent space to detect OoD, but is technically a model-dependent technique because the latent code is trained jointly with the downstream classification task.

One limitation of model-dependent OoD techniques is that they may discard information about $p(\mathrm{x})$ in learning the task-specific loss function $p_\theta(y|x)$. Consider a contrived binary classification model on images that learns to solve the task perfectly by discarding all information except the contents of the first pixel (no other information is preserved in the features). Subsequently, the model yields confident predictions on any distribution that happens to preserve identical first-pixel statistics. In contrast, density estimation in data space $x$ considers the structure of the entire input manifold, without bias towards a particular downstream task or task-specific compression.

In our work we estimate predictive uncertainty of the scoring model itself. Unlike predictive uncertainty methods applied to the task model's predictions, Generative Ensembles do not require task-specific labels to train. Furthermore, model-independent OoD detection aids interpretation of predictive uncertainty by isolating the uncertainty component arising from OoD inputs.

### 3.2 Adversarial Defense

Song et al. (2017) make the observation that adversarial examples designed to fool a downstream task have low likelihood under an independent generative model. They propose a "data purification" pipeline where inputs are first modified via gradient ascent on model likelihood, before passing it to the unmodified classifier. Their evaluations are restricted to $L^p$-norm attacks on in-distribution inputs to the task model, and do not take into account that the generative model itself may be susceptible to OoD errors. In fact, a preprocessing step with gradient ascent on model likelihood has the exact opposite of the desired effect when the input is OoD to begin with.

Our work considers adversarial defense in a broader OoD context. Although adversarial attacks literature typically considers small $L^p$-norm modifications to input (demonstrating the alarming sensitivity of neural networks), there is no such restriction in practice to the degree with which an input can be perturbed in a test setting. Adversarial defense is nothing more than making ML models robust to OoD inputs; whether they come from an attacker or not is irrelevant. We evaluate our methods on simple OoD transformations (flipping images), common ML datasets, and the adversaraial setting where a worst-case input is created from a single model in the ensemble.

He et al. (2017) demonstrate that ensembling adversarial defenses does not completely mitigate local sensitivity of neural networks. It is certainly plausible that sufficient search over a Generative

Ensemble's predictions can find OoD inputs with both low variance and high likelihood. The focus of our work is to measure the extent to which uncertainty estimation improves robustness to model mispecification error, not to present a provably secure system. Having said that, model-independent OoD detection is easy to obfuscate in a practical ML security setting since the user only has access to the task model. Furthermore, a Generative Ensemble's WAIC estimate can be made more robust by sampling additional models from the posterior over model parameters.

## 4 EXPERIMENTAL RESULTS

Following the experiments proposed by Liang et al. (2017) and Alemi et al. (2018b), we train OoD models on MNIST, Fashion MNIST, CIFAR-10 datasets, and evaluate anomaly detection on test samples from other datasets. In line with the aforementioned works, we measure anomaly detection capability based on AUROC over several quantities shown in Table 2. Our proposed quantities include single Wasserstein GAN (WGAN) discriminators (Arjovsky et al., 2017) with fine-tuning ($D$), ensemble variance of discriminators ($\mathrm{Var}(D)$), likelihood models ($\log p_\theta(\mathrm{x})$), and WAIC estimated using an ensemble of likelihood models. We follow the protocol as suggested by Lakshminarayanan et al. (2017) to use 5 independent models with different parameter initializations, trained on the full training set (no bootstrap). For likelihood estimators based on variational autoencoders (VAE), we also evaluate the rate term $D_{\mathrm{KL}}(q_\theta(z|x)\|p(z))$, which corresponds to information loss between the latent inference distribution and prior.

For MNIST and Fashion MNIST datasets, we use a VAE to predict a 16-sample Importance Weighted AutoEncoder (IWAE) bound. We extend the VAE example code[1] from Tensorflow Probability (Dillon et al., 2017) to use a Masked Autoregressive Flow prior (Papamakarios et al., 2017), and train the model for 5k steps. Additional architectural details are found in Appendix B.

Our WGAN model's generator and discriminator share the same architecture with the VAE decoder and encoder, respectively. The discriminator has an additional linear projection layer to its prediction of the Wasserstein metric. To ensure $D$ represents a meaningful discriminative boundary between the two distributions, we freeze the generator and fine-tune the discriminator for an additional 4k steps on stationary $p(\mathrm{x})$ and $q_\theta(\mathrm{x})$. We also include Gaussian noise adversarially perturbed by FGSM on a single model (Adversarial).

For CIFAR-10 WGAN experiments, we change the first filter size in the discriminator from 7 to 8. For log-likelihood estimation, we train a vanilla GLOW model (Kingma & Dhariwal, 2018) for 250k steps, as we require a more powerful generative model to obtain good results.

The baseline methods are model-dependent and learn from the joint distribution of images and labels, while our methods use only images. For the VIB baseline, we use the rate term as the threshold variable. The experiments in Alemi et al. (2018b) make use of (28, 28, 5) "location-aware" features concatenated to the model inputs, to assist in distinguishing spatial inhomogeneities in the data. In this work we train vanilla generative models with no special modifications, so for fair comparison we also train VIB without location-aware features. For CIFAR-10 experiments, we train VIB for 26 epochs and converge at 75.7% classification accuracy on the test set. All other experimental parameters for VIB are identical to those in Alemi et al. (2018b).

Despite being trained on strictly less data (no labels), our methods – in particular Generative Ensembles – outperform ODIN and VIB on most OoD tasks. The VAE rate term appears to be quite effective, outperforming likelihood and WAIC estimation in data space. It is robust to adversarial inputs on the same model, because the FGSM perturbation primarily minimizes the (larger) distortion component of the approximate likelihood. The performance of VAE rate versus VIB rate also suggests that latent codes learned from generative objectives are more useful for OoD detection that latent codes learned via a classification-specific objective.

### 4.1 FAILURE ANALYSIS

In this section we discuss the experiments in which Generative Ensembles performed poorly, and suggest simple fixes to address these issues.

---

[1] `https://github.com/tensorflow/probability/blob/master/tensorflow_probability/examples/vae.py`

Table 2: We train models on MNIST, Fashion MNIST, and CIFAR-10 and compare OoD classification ability to baseline methods using the threshold-independent Area Under ROC curve metric (AUROC). $D$ corresponds to single WGAN discriminators with 4k fine-tuning steps on stationary $p(\mathrm{x})$, $q(\mathrm{x})$. $\mathrm{Var}(D)$ is uncertainty estimated by an ensemble of discriminators. Rate is the $D_{\mathrm{KL}}$ term in the VAE objective. $\log p_\theta(\mathrm{x})$ is a single likelihood model (VAE, GLOW). WAIC is the Watanabe-Akaike Information Criterion as estimated by the Generative Ensemble. ODIN results reproduced from Liang et al. (2017). Best results for each task shown in bold.

| Train Dataset | OoD Dataset | ODIN | VIB | $D$ | $\mathrm{Var}(D)$ | Rate | $\log p_\theta(\mathrm{x})$ | WAIC |
|---|---|---|---|---|---|---|---|---|
| MNIST | Omniglot | **100** | 97.1 | 56.1 | 80.3 | 99.1 | 98.2 | **100** |
| | notMNIST | 98.2 | 98.6 | 93.1 | 99.6 | 99.9 | **100** | **100** |
| | Fashion MNIST | N/A | 85.0 | 83.1 | 99.9 | 94.7 | **100** | **100** |
| | Uniform | **100** | 76.6 | 95.6 | **100** | 99.3 | **100** | **100** |
| | Gaussian | **100** | 99.2 | 0.6 | **100** | **100** | **100** | **100** |
| | HFlip | N/A | 63.7 | 41.5 | 57.7 | **90.0** | 84.9 | 86.1 |
| | VFlip | N/A | 75.1 | 44.7 | 60.9 | **89.3** | 81.9 | 80.7 |
| | Adversarial | N/A | N/A | 30.8 | **100** | **100** | 0 | **100** |
| Fashion MNIST | Omniglot | N/A | 94.3 | 19.4 | 83.5 | **97.7** | 56.8 | 79.6 |
| | notMNIST | N/A | 89.6 | 22.3 | 96.0 | **99.7** | 92.0 | 98.7 |
| | MNIST | N/A | 94.1 | 70.1 | 74.7 | **97.1** | 42.3 | 76.6 |
| | Uniform | N/A | 79.6 | 0 | 82.7 | 95.6 | **100** | **100** |
| | Gaussian | N/A | 89.3 | 0 | 99.8 | 89.2 | **100** | **100** |
| | HFlip | N/A | 66.7 | 58.0 | 54.1 | **72.4** | 59.4 | 62.3 |
| | VFlip | N/A | **90.2** | 69.6 | 69.6 | 87.1 | 66.8 | 74.0 |
| | Adversarial | N/A | N/A | 0 | **100** | **100** | 0 | **100** |
| CIFAR-10 | CelebA | N/A | 73.5 | 56.5 | 74.3 | N/A | 99.7 | **99.9** |
| | SVHN | N/A | 52.8 | 68.9 | 61.4 | N/A | 7.5 | **100** |
| | ImageNet32 | 81.6 | 70.1 | 47.1 | 62.9 | N/A | 93.8 | **95.6** |
| | Uniform | 99.2 | 54.0 | **100** | **100** | N/A | **100** | **100** |
| | Gaussian | 99.7 | 45.8 | **100** | **100** | N/A | **100** | **100** |
| | HFlip | N/A | 50.6 | **52.0** | 50.3 | N/A | 50.1 | 50.0 |
| | VFlip | N/A | 51.2 | **60.9** | 52.3 | N/A | 50.6 | 50.4 |

In an earlier draft of this work, a VAE trained on Fashion MNIST performed poorly on all OoD datasets when using $\log p_\theta(\mathrm{x})$ and WAIC metrics. This was surprising, since the same metrics performed well when the same VAE architecture was trained on MNIST. To explain this phenomenon, we show in Figure 3 inputs and VAE-decoded outputs from Fashion MNIST and MNIST test sets. Fashion MNIST images are reconstructed properly, while MNIST images are are barely recognizable after decoding.

A VAEs training objective can be interpreted as the sum of a pixel-wise autoencoding loss (distortion) and a "semantic" loss (rate). Even though Fashion MNIST appears to be better reconstructed in a semantic sense, the distortion values between the FMNIST and MNIST test datasets are numerically quite similar, as shown in Figure 3. Distortion terms make up the bulk of the IWAE predictions in our models, thus explaining why $\log p_\theta(\mathrm{x})$ was not very discriminative when classifying OoD MNIST examples.

Higgins et al. (2016) propose $\beta$-VAE, a simple modification to the standard VAE objective: $p(x|z) + \beta \cdot D_{\mathrm{KL}}(q_\theta(z|x)\|p(z))$. $\beta$ controls the relative balance between rate and distortion terms during training. Setting $\beta < 1$ is a commonly prescribed fix for encouraging VAEs to approach the "autoencoding limit" and avoid posterior collapse (Alemi et al., 2018a). At test time, this results in higher-fidelity autoencoding at the expense of higher rates, which seems to be a more useful signal for identifying outliers than the total pixel distortion (also suggested by Table 2, column 7). Re-training the ensemble with $\beta = .1$ encourages a higher distortion penalty during training, and thereby fixes the OoD detection model.

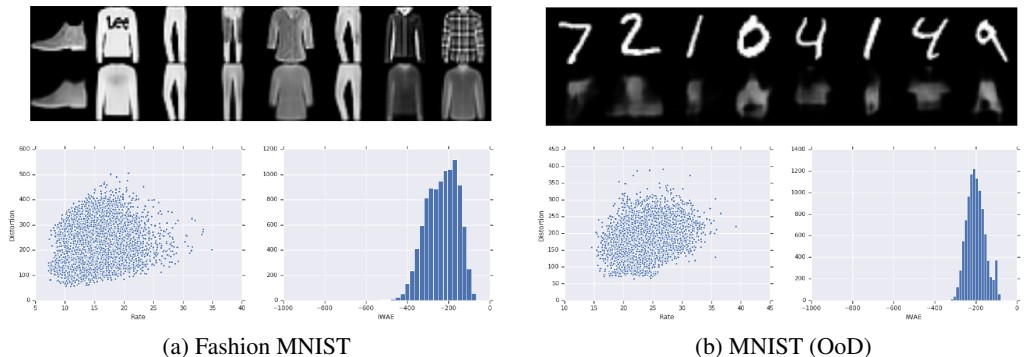

(a) Fashion MNIST          (b) MNIST (OoD)

Figure 3: Top: Inputs and decoded outputs from a VAE trained on Fashion MNIST($\beta = 1$) for Fashion MNIST (left) and MNIST (right). Although Fashion MNIST inputs appear to be better reconstructed (suggesting higher likelihoods), they have comparable distortions to MNIST. The bottom row shows that Fashion MNIST and MNIST test samples have comparable rate-distortion scatter plots and IWAE histograms.

## 4.2   CREDIT CARD ANOMALY DETECTION

We consider the problem of detecting fraudulent credit card transactions from the Kaggle Credit Fraud Challenge (Dal Pozzolo et al., 2015). A conventional approach to fraud detection is to include a small fraction of fraudulent transactions in the training set, and then learn a discriminative classifier. Instead, we treat fraud detection as an anomaly detection problem where a generative model only sees normal credit card transactions at training time. This is motivated by realistic test scenarios, where an adversary is hardly restricted to generating data identically distributed to the training set.

We compare single likelihood models (16-sample IWAE) and Generative Ensembles (ensemble variance of IWAE) to a binary classifier baseline that has access to a training set of fraudulent transactions in Table 3. The classifier baseline is a fully-connected network with 2 hidden ReLU layers of 512 units, and is trained using a weighted sigmoid cross entropy loss (positive weight=580) with Dropout and RMSProp ($\alpha = 1e-5$). The VAE encoder and decoder are fully connected networks with single hidden layers (32 and 30 units, respectively) and trained using Adam ($\alpha = 1e-3$).

Unsurprisingly, the classifier baseline performs best because fraudulent test samples are distributed identically to fraudulent training samples. Even so, the single-model density estimation and Generative Ensemble achieve reasonable results.

Table 3: Comparison of density-based anomaly detection approaches to a classification baseline on the Kaggle Credit Card Fraud Dataset. The test set consists of 492 fraudulent transactions and 492 normal transactions. Threshold-independent metrics include False Positives at 95% True Positives (FPR@95%TPR), Area Under ROC (AUROC), and Average Precision (AP). Density-based models (Single IWAE, WAIC) are trained only on normal credit card transactions, while the classifier is trained on normal and fraudulent transactions. Arrows denote the direction of better scores.

| Method | FPR@95%TPR ↓ | AUROC ↑ | AP ↑ |
|---|---|---|---|
| Classifier | 4.0 | 99.1 | 99.3 |
| Single IWAE | 15.7 | 94.6 | 92.0 |
| WAIC | 15.2 | 94.7 | 92.1 |

## 5 DISCUSSION AND FUTURE WORK

OoD detection is a critical piece of infrastructure for ML applications where the test data distribution is not known at training time. We present Generative Ensembles, a simple yet powerful technique for model-independent OoD detection that improves density models with uncertainty estimation.

An important future direction of research is that of *scalability*: learning good generative models of semantically rich, high-dimensional inputs (e.g. video) is an active research area in its own right. An open question is whether an ensemble of weak generative models (where each model may not necessarily generate high-quality samples) can still yield density and uncertainty predictions useful enough for OoD detection. Preliminary evidence on CIFAR-10 are promising; although the ensemble average on the test set is $\sim 3.5$ bits/dim and samples from the prior do not resemble any recognizable objects, the ensemble still performs well at OoD detection. In future work we will explore other methods of de-correlating samples from the posterior over model parameters, as well as combining independent scores ($D$, Rate, $\log_\theta p(\mathrm{x})$, WAIC) into a more powerful OoD model.

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

## A    TERMINOLOGY AND ABBREVIATIONS

| | |
|---|---|
| $p(\mathrm{x})$ | Training data distribution |
| $q(\mathrm{x})$ | OoD data distribution |
| $p_\theta(\mathrm{x})$ | Learned approximation of true distribution with parameters $\theta$. May be implicitly specified, i.e. a fully-observed density model |
| $q_\theta(\mathrm{x})$ | Learned approximation of OoD distribution with parameters $\theta$. May be implicitly specified, i.e. via a GAN discriminator that learns $p(x)/q_\theta(x)$ |
| OoD Input | Out-of-Distribution Input. Invalid input to a ML model |
| Anomaly | Synonym with OoD Input |
| Epistemic Uncertainty | Variance in a model's predictive distribution arising from ignorance of true model parameters for a given input |
| Aleatoric Uncertainty | Variance in a model's predictive distribution arising from inherent, irreducible noise in the inputs |
| Predictive Uncertainty | Variance of a model's predictive distribution, which takes into account all of the above |
| MNIST | Dataset of handwritten digits (size: 28x28) |
| FashionMNIST | Dataset of clothing thumbnails (size: 28x28) |
| CIFAR-10 | Dataset of color images (size: 32x32x3) |
| GAN | Generative Adversarial Network. See Goodfellow et al. (2014) |
| FSGM | Fast Sign Gradient Method |
| WGAN | Wasserstein GAN. See Arjovsky et al. (2017) |
| VAE | Variational Autoencoder. See Kingma & Welling (2013); Rezende et al. (2014) |
| Rate | $D_{\mathrm{KL}}(q_\theta(z|x)\|p(z))$ term in the VAE objective. Information loss between encoder distribution and prior over latent code |
| IWAE | Importance Weighted Autoencoder |
| GLOW | A generative model based on normalizing flows. See Kingma & Dhariwal (2018) |
| ODIN | Out-of-DIstribution detector for Neural networks. See Liang et al. (2017) |
| VIB | Variational Information Bottleneck. See Alemi et al. (2018b) |
| WAIC | Watanabe-Akaike Information Criterion. See Watanabe (2010) |
| AUROC | Area Under ROC Curve |
| FPR@95%TPR | False Positives at 95% True Positives |
| AP | Average Precision |

## B    VAE ARCHITECTURAL DETAILS

We use a flexible learned prior $p_\theta(\mathrm{z})$ in our VAE experiments, but did not observe a significant performance difference compared to the default mixture prior in the base VAE code sample. We use an alternating chain of 6 MAF bijectors and 6 random permutation bijectors. Each MAF bijector uses TensorFlow Probability's default implementation with the following parameter:

```
shift_and_log_scale_fn=tfb.masked_autoregressive_default_template(
    hidden_layers=(512, 512))
```

Models are trained with Adam ($\alpha = 1\mathrm{e}{-3}$) with cosine decay on learning rate.

