# OpenReview forum: "Generative Ensembles for Robust Anomaly Detection"
_ICLR.cc/2019/Conference_

### Official Review · AnonReviewer2 · 2018-11-01
**Interesting combination of the previous work with useful results.**

**Rating:** 6
**Confidence:** 3

**Review:**

The authors present an OOD detection scheme with an ensemble of generative models. When the exact likelihood is available from the generative model, the authors approximate the WAIC score. For GAN models, the authors compute the variance over the discriminators for any given input. They show that this method outperforms ODIN and VIB on image datasets and also achieves comparable performance on Kaggle Credit Fraud dataset.

The paper is overall well-written and easy to follow. I only have a few comments about the work.

I think the authors should address the following points in the paper.
- What is the size of the ensemble for the experiments?
- How does the size of the ensemble influence the measured performance?
- It is Fast Gradient Sign Method (FGSM), not FSGM. See [1]. Citing [1] for FGSM would also be appropriate.

Quality. The submission is technically sound. The empirical results support the claims, and the authors discuss the failure cases.
Clarity. The paper is well-written and easy to follow while providing useful insight and connecting previous work to the subject of study.
Originality. To the best my knowledge, the proposed approach is a novel combination of well-known techniques.
Significance. The presented idea improves over the state-of-the-art.


References
[1] I. Goodfellow, J. Shlens, and C. Szegedy, “Explaining and Harnessing Adversarial Examples,” in ICLR, 2015.
-------------------
Revision. The rating revised to 6 after the discussion and rebuttal.

---

> ### Author Response · Authors · 2018-11-19
> **Thanks!**
>
> We thank Reviewer 3 for the review and highlighting missing details from our paper. We’ve added them into the paper.
>
> > - How does the size of the ensemble influence the measured performance?
>
> For CIFAR10, we have found 5 ensembles to make a large difference over 3 ensembles (about .7 AUROC). There seem to be diminishing returns for models > 5.
>
> > - It is Fast Gradient Sign Method (FGSM), not FSGM. See [1]. Citing [1] for FGSM would also be appropriate.
>
> Fixed, and already cited. Thanks!

---

### Official Review · AnonReviewer3 · 2018-11-01
**Well below the ICLR level**

**Rating:** 4
**Confidence:** 5

**Review:**

- Novelty is minimal and is well below the level required by ICLR.

- The reasoning lists the problems of GANs and then the fact that GAN ensembles would target that, based on a toy example in Figure 2.

- Why to choose GANs though in the first place? Given the buildup, and given the other well-known training issues about GANs, are they the right choice for the basic modeling units, i.e. the ensemble units, in such case? A GANs adversary bases its comparisons on individual data points, rather than on distribution comparisons or on groups of points like MMD, etc. I understand the reasoning behind the choice of generative models (GMs), but it is choosing GANs out of the set of GMs in this particular case that I am referring to.

- The paper is quite well written. The ideas as well as the reasoning flow very smoothly.

- Experiments are well prepared.

Rather minor:
- page 1: "When training and test distributions differ, neural networks may provide ..." This is true but may be a clarification here regarding the fact that the neural networks involved with several modeling problems, e.g. the ones trained for domain adaptation or meta-learning tasks, target this shift or difference in domains, and typically provide a way to tackle this problem.



Uodate: Read the rebuttal. My score remains unchanged.

---

> ### Author Response · Authors · 2018-11-19
> **Addressing concerns about novelty and use of GANs**
>
> We thank Reviewer 2 for their praise and raising concerns about novelty. It is an important point worth discussing.
>
> In addition to proposing a superior method for anomaly detection, part of the novel contribution in this work involved synthesizing concepts from multiple fields likelihood estimation techniques from deep generative models, adversarial defense, model uncertainty, challenging discriminative anomaly detection methods and their relationship to GAN discriminators.
>
> We tie these disparate concepts together into a unified perspective on the OoD problem. Therefore, we took great care into making sure the motivation of our work transitions smoothly, perhaps even to the point of stating the obvious to Reviewer 2. We emphasize that to our knowledge, our work is the first to extend our understanding of the OoD problem in context of prior work in generative modeling, Bayesian Deep Learning, and anomaly detection applications for modern generative models. These connections are not well known in the community and we hope that our paper will amend that.
>
> Additional novel aspects of this work: The observation that density estimators (as implemented by a deep generative model) are NOT robust to OoD inputs themselves is a novel observation, concurrent with another ICLR submission. To our knowledge, we are also the first work to leverage the modern advancements in deep generative models to perform anomaly detection on high-dimensional inputs such as images.
>
> To address R2’s comments “The reasoning lists the problems of GANs” and “Why to choose GANs though in the first place?”, we emphasize that we are not saying GANs shouldn’t be used for anomaly detection, only that their lack of exact likelihoods presents some challenges. We make an effort to make them work in our paper in our comparison to other generative model families.
>
> > - page 1: "When training and test distributions differ, neural networks may provide ..."
>
> There are varying degrees of “out-of-distribution-ness” at test time. One way to carve up the problem specification is to consider inputs that (1) are different than the training set but you want the model to perform well on anyway, e.g. a subtle change in physics parameters a robot encounters when deployed. (2) inputs the model has no business classifying, i.e. showing a picture of a building to a cat/dog classifier.
>
> The first situation is what you are describing, in which methods like sim2real, domain adaptation, meta-learning can address. As we stated in Section 3.1, our paper primarily deals with the second case, in which you don’t want the model to give bogus outputs for bogus inputs, which also may be adversarial. We appreciate the feedback that this might be confusing if the reader is assuming problem formulation (1); we welcome the other reviewers to chime in here if it would make things more clear to state this.

---

### Official Review · AnonReviewer1 · 2018-11-03
**Needs a lot of work on improving technical rigor and clarity**

**Rating:** 5
**Confidence:** 4

**Review:**

Note to Area Chair: Another paper submitted to ICLR under the title “Do Deep Generative Models Know What They Don’t Know?” shares several similarities with the current submission.

This paper highlights a deficiency of current generative models in detecting out-of-distribution based samples based on likelihoods assigned by the model (in cases where the likelihoods are well-defined) or the discriminator distribution for GANs (where likelihoods are typically not defined). To remedy this deficiency, the paper proposes to use ensembles of generative models to obtain a robust WAIC criteria for anomaly detection.

My main concern is with the level of technical rigor of this work. Much of this has to do with the presentation, which reads to me more like a summary blog post rather than a technical paper.
- I couldn’t find a formal specification of the anomaly detection setup and how generative models are used for this task anywhere in the paper.
- Section 2 seems to be the major contribution of this work. But it was very hard to understand what exactly is going on. What is the notation for the generative distribution? Introduction uses p_theta. Page 2, Paragraph 1 uses q_theta (x). Eq. (1) uses p_theta and then the following paragraphs use q_theta.
- In Eq. (1), is theta a random variable?
- How are generative ensembles trained?  All the paper says is “independently trained”. Is the parameter initialization different? Is the dataset shuffling different? Is the dataset sampled with replacement (as in bootstrapping)?
- “By training an ensemble of GANs we can estimate the posterior distribution over model deciscion boundaries D_theta(x), or equivalently, the posterior distribution over alternate distributions q_theta. In other words, we can use uncertainty estimation on randomly sampled discriminators to de-correlate the OoD classification errors made by a single discriminator” Why is the discriminator parameterized by theta? What is an ensemble of GANs? Multiple generators or multiple discriminators or both? What are “randomly sampled discriminators”? What do the authors mean by "posterior distribution over alternate distributions"?

With regards to the technical assessment, I have the following questions for the authors:
- In Figure 1, how do the histograms look for the training distribution of CIFAR? If the histograms for train and test have an overlap much higher than the overlap between the train of CIFAR and test set of any other distribution, then ensembling seems unnecessary and anomaly detecting can simply be done via setting a maximum and a minimum threshold on the likelihood for a test point. In addition to the histograms, I'd be curious to see results with this baseline mechanism.
- Why should the WAIC criteria weigh the mean and variance equally?
- Did the authors actually try to fix the posterior collapse issue in Figure 3b using beta-VAEs as recommended? Given the simplicity of implementing beta-VAEs, this should be a rather easy experiment to include.

Minor typos:
- ODIN and VIB are not defined in the abstract
- Page 3: “deciscion”
- Page 2, para 2: “log_\theta p(x)”

---

> ### Author Response · Authors · 2018-11-19
> **Addressed issues of technical clarity, performed follow-up experiments on posterior collapse**
>
> Thank you for the detailed review and critique.
>
> We agree that “Do Deep ... They Don’t Know?” shares a concurrent discovery with us in identifying how generative models assign wrong likelihood to OoD inputs, and have updated our paper to cite their contribution. Our contributions differ in that their work performs analysis of why this phenomenon occurs, while we demonstrate that this can be fixed by using uncertainty estimation and WAIC, and then apply these fixed models to the OoD problem.
>
> We agree that our paper could use more technical clarity, i.e. make this work easier to reproduce. The open-sourced code will be linked to the paper after double-blind review process, which we believe to be the highest standard of technical clarity when specifying our method and evaluation metrics. In the meantime, we’ve also done the following:
>
> 1. We’ve clarified Section 4 to re-iterate that our anomaly detection problem specification is identical to that of Liang et al. 2017 and Alemi et al. 2017, and our evaluation metric (AUROC) is the same.
>
> 2. Clarified the notation of our notation for p, q, p_theta, q_theta in the paper. We think that R1’s confusion on our GAN ensemble setup can be addressed by clarifying the reasoning behind our terminology, and explaining a bit further what it means to “randomly sample a discriminator from a posterior distribution over alternate distributions”
>
> The choice of terminology is motivated by our GAN variant of generative ensembles. If p(x) is the true generative distribution, p_theta(x) is some generative model’s approximation of it. In Eq (1), theta is a (multivariate) random variable parameterizing an abstract generative model (e.g. weights in a neural network). We’ve clarified this in the intro.
>
> In the case of GANs, a subset of the variable theta parameterizes the generator and a subset of theta parameterizes the discriminator. Therefore, samples from the generator come from a generative distribution q_\theta(x). We notate a GAN generator’s distribution as q_\theta(x) and not p_\theta(x) (which we use for referring to normalizing flow and VAE likelihood models) is that in GANs, the discriminator is being optimized to learn a likelihood ratio p(x) / q_\theta(x). That is, separating true data samples from p(x) from OoD samples from q_\theta(x).
>
> Thus, q(x) and q_theta(x) always refer to OoD distributions. This also makes discussion more clear in the context of discriminative anomaly detection classifiers (which learn p(x)/q(x)) and GAN discriminators (which learn p(x)/q_theta(x)).
>
> In Section 2.1, we mention “randomly sampled discriminators” and “posterior distribution over alternate distributions”. Models (theta) trained under SGD can be assumed to be drawn randomly from some posterior distribution over p(theta|x). In a GAN, random variable theta specifies the  alternate distribution q_\theta(x), or equivalently, the implicit discriminator likelihood ratio p(x) / q_\theta(x) (when the discriminator is trained with sigmoid cross entropy, which we do). Our GAN ensembles samples entire GANs (i.e. generator and discriminator) together, by training 5 GANs independently and then combining discriminator predictions for OoD classification. It would be problematic to sample only discriminators in the training process, since that does not change q_\theta(x) (and there is the question of how feedback to the generators should be accomplished in this manner).
>
> Technical assessment questions:
>
> - Re: Histograms. This is a reasonable suggestion, and resembles the interpretation of likelihood predictions as a feature, rather than a scoring function. The scoring function you propose is a min/max function over the distribution of features. Another approach would be a statistical hyppothesis test using the training distribution’s likelihood predictions as the variable of interest. Unfortunately, the likelihoods of OoD distributions often overlap with the in-distribution test samples (MNIST and Fashion MNIST VAEs). In training a GLOW model, you will also find a gap between train and test likelihoods. So generative models are not good enough yet to reduce the generalization gap of likelihood models zero.
>
> - We refer the reviewer to "Understanding predictive information criteria for Bayesian models" (Gelman et al.) for a motivation of the WAIC objective. In short, the variance term is a correction for how much the fitting of k parameters will increase predictive accuracy, by chance alone. K is estimated by the variance.
>
> - Re: Posterior Collapse: Good suggestion! We went back to our VAE setup and ran a few follow-up experiments to prove this hypothesis. The short answer is that “yes, decreasing Beta reduced posterior collapse and made things better”. We’ve edited section 4.1 to document our findings.
>
> Minor typos: They have been fixed in the latest revision. Thank you so much for catching these!

---

> > ### Comment · AnonReviewer1 · 2018-12-06
> > **Response**
> >
> > Thanks for the response!
> >
> > - Some of my concerns regarding clarity have been addressed. Must note that clarity can still benefit from some more editing (a self-contained paper on anomaly detection will describe the experimental setup rather than just referring the reader to two other papers, the GAN notation of q_\theta is clear to me now but is frankly unnecessary imo, details on how many ensembles were trained and how did they differ, etc.).
> >
> > - Re: Posterior Collapse. I also appreciate the results on this experiment.
> >
> > Based on the first two points, I have updated by score. However, I found the response to the other concerns rather dissatisfying.
> >
> > - Re: Histograms. Besides including the training likelihood results for the datasets in the submission, I think the AUROC based on an "overlap" based scoring rule is a very reasonable and important baseline to include before the expensive process of training ensembles.
> > - WAIC. I think my question was orthogonal to the link you provide to. I was more interested in knowing why the mean and variance terms should be weighted equally, rather than having a hyperparameter controlling their strengths which could be decided based on e.g., a validation set. Some intuition/experiments in this regard would have been welcome.

---

> > > ### Author Response · Authors · 2018-12-06
> > > **Re: Response**
> > >
> > > Thank you for the detailed feedback. That's really helpful.
> > >
> > > Re: Histograms. We are a bit confused now as to what you mean by "overlap" based scoring rule. Under our experimental setup, anomaly detection is performed on a per-example basis from the test distribution. Although we eval AUROC on an empirical test set, we don't have access to the a population of test samples  in  the scoring rule. So it is not possible to compute "overlap" between two histograms of test points because we evaluate each test point independently.
> > >
> > > Referencing your earlier comment, it is possible to use training histograms to build a scoring rule. As you described, we can construct indicator function that classifies a data point as an anomaly if it has lower likelihood than the least probable training point or higher likelihood than the most probable training point (where training points are from the empirical training distribution). We can update our results with such a baseline (if this is what you intended), though as we've said before, MNIST and Fashion MNIST test distributions (and NotMNIST too) have considerably overlapping histograms, so it is doomed to fail. We don't think that this is a sufficiently strong baseline for the purposes of evaluating our method.
> > >
> > > - Re: As the number of data points n grows large, the Expectation of WAIC converges to generalization loss (which is a surrogate objective for KL distance between model and true distribution). See Eq 31. http://www.jmlr.org/papers/volume14/watanabe13a/watanabe13a.pdf and Watanabe 2009, 2010b for proofs. Now suppose we have a modified objective WAIC2 = log p(x) + alpha * Var[log p(x)]. For alpha != 1, this would result in a biased asymptotic estimate of generalization error.
> > >
> > > That said, calibrating alpha according to a validation set might yield better AUROC. However, we avoid doing this in our experiments because it would presupposing an OoD distribution (validation set), which may lead to poor performance on the test OoD distribution (which may be different than validation set). Also, to making comparison to prior work easier (since AUROC is supposed to be threshold-independent for a single scalar), we didn't modify the WAIC score function.

---

### Author Response · Authors · 2018-11-19
**Overall rebuttal comment from authors**

We thank the reviewers for helpful feedback and highlighting points of confusion in our paper.

In considering all 3 reviewers’ comments (R1 “reads more like a summary blog post”, R2 “the ideas as well as reasoning flow smoothly”, R3 “well-written and easy to follow while providing useful insight and connecting previous work to the subject of study”) we believe that all reviewers consider our presentation to be logically clear, but may be lacking in technical clarity (raised by Reviewer 1) or novelty (raised by Reviewer 2). There is especially some confusion regarding our notation and how it relates to GAN models for anomaly detection (e.g. “posterior distribution over alternate distributions”).

To address technical clarity issues raised by R1, we’ve answered their questions in comments and made edits to our paper to make the problem setup and notation more clear.  We’ve responded directly to R2’s comment on why we believe our work is novel.

Finally, we’ve updated the paper with improved VAE experiments on Fashion MNIST (confirming our hypothesis of posterior collapse).

---

> ### Author Response · Authors · 2018-11-26
> **Updated Fashion MNIST numbers to fix a bug.**
>
> Our improved VAE experiments on Fashion MNIST had a minor evaluation bug in which some OoD test samples from Omniglot got mixed up into other distributions' evaluation. We've updated the paper to fix this error. After the rebuttal deadline, we'll update our related work section to discuss some of the GAN papers R2 mentioned in their recent comment.

---

### Meta-Review · Area_Chair1 · 2018-12-14
**Promising but more work needed to reach maturity**

**Confidence:** 4
**Recommendation:** Reject

**Metareview:**

This paper suggests the use of generative ensembles for detecting out-of-distribution samples.

The reviewers found the paper easy to read, especially after the changes made during the rebuttal. However, further elaboration in the technical descriptions (and assumptions made) could make the work seem more mature, as R2 and R1 point out.

The general feeling by reading the reviews and discussions is that this is promising work that, nevertheless, needs some more novel elements. A possible avenue for increasing the contribution of the paper is to follow R1’s advice to extract more convincing insights from the results.